# Watching the Russian–Ukrainian War: Comparison Between Europe and North America

**DOI:** 10.3390/bs15101319

**Published:** 2025-09-26

**Authors:** Esther R. Greenglass, Petra Begic, Petra Buchwald, Taina Hintsa, Krzysztof Kaniasty, Petri Karkkola, Iva Poláčková Šolcová

**Affiliations:** 1Department of Psychology, York University, Toronto, ON M3J 1P3, Canada; 2Department of Psychology, University of Wuppertal, 42119 Wuppertal, Germany; petrabegic@gmx.de (P.B.); pbuchw@uni-wuppertal.de (P.B.); 3Department of Educational Sciences and Psychology, University of Eastern Finland, 80100 Joensuu, Finland; taina.hintsa@uef.fi (T.H.); petri.karkkola@uef.fi (P.K.); 4Department of Psychology, Indiana University of Pennsylvania, Indiana, PA 15701, USA; kaniasty@iup.edu; 5Polish Academy of Sciences, 1 Defilad Square, 00-901 Warsaw, Poland; 6Institute of Psychology, The Czech Academy of Sciences & Faculty of Humanities, Charles University in Prague, 128 43 Prague, Czech Republic; polackova@praha.psu.cas.cz

**Keywords:** war, anxiety, denial

## Abstract

Reports indicate that millions of people have been watching the Russian–Ukrainian war that broke out on 24 February 2022. This research studies the relationship between watching the war and psychological reactions in 1260 university students who responded to an online questionnaire related to watching the war on various media forms. Data were collected from April to October 2022 from five national samples from Europe (Germany, Finland, and the Czech Republic) and North America (Canada and the U.S.). Since European countries are assumed to have greater ties with the countries at war, anxiety, anger, and denial while watching the war should be greater in European participants than in North American ones. Worry about the war should be greater when more hours are spent watching the war, and anxiety related to the war should decrease with self-efficacy. ANOVA results showed that European participants spent more hours watching the war, worried more, and experienced greater distress than North American ones. Path analysis showed that having relatives, friends, or colleagues in Ukraine or Russia was associated with worry about the war through hours spent watching it. Self-efficacy was negatively related to anxiety. Psychological distress related to watching the war was far-reaching, extending to countries beyond Ukraine and Russia.

## 1. Introduction

The unprovoked Russian attack against Ukraine on 24 February 2022 resulted in an extensive loss of life of citizens and military personnel alike, as well as the destruction of cities and towns, buildings, and infrastructure. Reports of the ongoing war indicate that millions of people worldwide have been watching the war on various media forms since it began. On 7 March 2022, TikTok videos tagged with #ukrainewar were viewed more than 600 million times, and in 2022, almost 180,000 Instagram posts were made ([38]).

Considerable research has been conducted on the psychological effects of watching violence, including war, on TV and social media. It is possible that individuals may experience negative emotions while watching war scenes on social media without being physically present during the hostilities. Emotional suffering related to a war may occur not only due to direct exposure but also through indirect sources, such as viewing war scenes via television or social media. Taken together, research indicates that watching violence on TV or social media can induce negative psychological feelings and distress in viewers ([16]; [28]). In past research, television exposure to 9/11 and psychological symptomatology were positively related, as shown in cross-sectional studies among New York residents ([5]). The effects of the 9/11 events were seen in the U.K., thousands of miles away, for 6 months after 9/11 ([18]). These findings parallel the ideas put forth by cultivation theory ([13]), which focuses on the influence of television on viewers. Research findings suggest that cultivation can take place through both direct and indirect processes, as demonstrated by [7] ([7]), who found that parents who watch more programs portraying crime and violence are more likely to warn their children about crime during their high school years; these warnings, in turn, increased the students’ own crime estimates. Taken together, research suggests that media messages can affect one’s cognitions as well as one’s emotional reactions to media portrayals of violence.

According to [27] ([27]), stressors are indirectly related to emotional outcomes, such as anxiety, through cognitive appraisal. Specifically, stressors are primarily appraised by evaluating whether or not a stressor has the potential to cause harm or loss (i.e., threat). If stressors are appraised as threatening, they are then secondarily appraised through an evaluation of one’s resources to manage the stressor (i.e., coping). If one perceives that the stressor is threatening and that the demands of the encounter exceed one’s resources to overcome the stressor, negative emotions (e.g., anxiety) ensue. Traditionally, research on coping has distinguished between problem-focused coping, or instrumental approaches to coping, and emotion-focused coping. Problem-focused coping consists of efforts aimed at altering the person–environment transaction or altering or managing the source of the stress, while emotion-focused coping is focused on regulating emotional responses elicited by the situation ([12]). This may include avoidance, denial, and wishful thinking, often referred to as disengagement coping ([8]). Emotion-focused coping strategies, such as denial, i.e., when one refuses to believe that the problem is real, are predominantly used in situations that cannot be altered ([27]). Denial is a maladaptive coping strategy ([37]) and can be a strong predictor of distress ([1]). Generally, denial is ineffective in reducing distress as it does nothing to manage the threat and its impact. It is most commonly used when an individual confronts an uncontrollable stressor.

### 1.1. The Present Research

The present research examines psychological reactions to the Russian–Ukrainian war in North American and European university students who were watching the war on various media platforms. In light of the widespread interest in the progress of the war internationally, and given the research showing the deleterious effects of violence depictions, it was important to investigate whether psychological reactions to watching the war varied with the viewer’s continental location. Specifically, the present research examines the relationship between the reported number of hours spent watching the war on social media and self-reported distress in 1263 university students residing in Europe and North America. Specifically, the European sample was taken from three countries: Germany, Finland, and the Czech Republic. The North American sample comprised participants from the U.S. and Canada. According to previous research, anxiety, anger, and worry often covary and are considered symptoms of psychological distress ([15]).

In this study, we were interested in the psychological effects of immersion in the war through social media. We used the number of hours spent watching the war (on TV and/or social media) to represent the extent of engagement with watching the war. In this study, we did not differentiate between different media forms. We measured media exposure by aggregating TV and social media into a single item. It was expected that the number of hours spent watching the war would be greater in European individuals than in North American individuals ([9]). Furthermore, it was hypothesized that psychological distress while watching the war would be greater in European viewers than in their North American counterparts for the following three reasons: First, greater distress was expected in individuals who reside in countries physically closer to the war (i.e., European countries) compared to North American countries (i.e., the U.S. and Canada). As suggested by [25] ([25]), the closer their country of residence is to the war, the more likely that people may be negatively affected by depictions of violence. Second, viewers should report greater psychological distress if they report having more relatives or friends currently living in Ukraine or Russia. Worry and distress were expected to be associated with concern for the well-being of relatives and friends who were directly affected by the war. Furthermore, European participants were expected to have more relatives in the war zone than their North American counterparts. Having relatives in the war zone should be associated with more hours spent watching the war and with greater worry about the war ([29]). Therefore, European participants should report spending more hours watching the war and should experience greater negative emotional reactions. Third, the more viewers identify with the victims of the violence of war and the greater the perceived similarity between the viewers and the victims, the more distress should be reported by the viewers. Perceived similarity to the victims of violence may be associated with the viewer’s distress symptoms ([17]). Since Europeans have had closer relationships with individuals in either Russia or Ukraine and likely have more relatives living close to the location of the war, they will likely watch the war more frequently and identify more closely with the victims of the violence. Therefore, Europeans should experience more distress, defined as anxiety, anger, and worry, when watching the war compared to North Americans.

### 1.2. Individual Differences, Anxiety, and Worry

Research has shown that individuals’ anxiety and worry are significantly related to gender, self-efficacy, beliefs in their ability to control their functioning, and events in their lives ([4]). Findings show that individuals who report high scores of self-efficacy are more likely to respond to stressors with less worry as well as lower levels of anxiety. In their research, [35] ([35]) examined the relationship between self-efficacy, job stress, and burnout in teachers. The results showed that highly self-efficacious teachers perceived the objective demands of daily teaching as less threatening than teachers with lower self-efficacy. The concept of self-efficacy aligns with the concept of salutogenesis ([2]), which is often applied to health promotion and refers to an orientation towards problem-solving and finding solutions to challenges that increase one’s health and well-being. In the present study, it is expected that the higher the self-efficacy, the lower the anxiety and worry about the war. Women should express greater worry and anxiety related to the war in light of previous research findings showing that women often express greater anxiety compared to men ([30]).

### 1.3. Theoretical Model

Theoretically, stress appraisal can be invoked to describe the relationship between a stressor (e.g., watching the war) and negative reactions. According to Lazarus and Folkman, “Psychological stress results from a particular relationship between the person and the environment that is appraised by the person as taxing or exceeding his or her resources and endangering their well-being” (p. 19). According to [27] ([27]), stress is rooted in emotions and arises largely as a result of an individual’s cognitive appraisal of a situation. With the primary appraisal, a stressor, for example, watching the war, is evaluated regarding its significance to the individual. In particular, an event is evaluated according to whether it can cause a threat or danger. If so, options for coping are evaluated in the secondary appraisal by evaluating one’s resources. In line with Transactional Stress Theory ([27]), one may distinguish between problem-focused and emotion-focused coping styles. While the former aims at altering the stressor, the latter is focused on managing one’s emotional states in response to the stressor. In general, escape, avoidance, and denial are coping strategies commonly observed in situations where no solution is seen as possible. On the other hand, emotion-based coping strategies are predominantly used to manage one’s emotional reactions to a stressor and are often associated with distress ([27]).

Here, we present a theoretical model that integrates the number of hours spent watching the war with having relatives in Ukraine or Russia, emotional factors, such as anxiety, anger, and worry about the war, denial (which is a maladaptive coping strategy), gender, and self-efficacy (see Figure 1). The model was tested with university students in Europe and North America in order to compare psychological reactions to watching the war in countries that varied predominantly in (physical) distance from the hostilities. Three countries encompassed the European sample (Finland, Germany, and the Czech Republic), while the U.S. and Canada comprised the North American sample.


**Hypotheses.**


In this study, the number of hours spent watching the war was a proxy for involvement in the war. The results of previous research suggest that the extent of exposure to armed conflict through media is associated with the degree of psychological distress and post-traumatic symptoms ([11]). Thus, in the present study, it was hypothesized that the more hours spent watching the war, the greater the reported distress. It was also hypothesized that European participants would spend more hours watching the war than their North American counterparts and that the former would experience more distress. These hypotheses follow previous research showing that the closer their country of residence is to the war, the more likely people will be negatively affected by depictions of violence ([25]). Perceived similarity to the victims of violence may also be associated with the viewer’s distress symptoms ([17]). Since Europeans likely perceive themselves as being more similar to the victims of the violence in Ukraine than North Americans, they would be expected to report greater distress. It was further hypothesized that the more relatives or friends that participants had in Ukraine or Russia, the more hours they would report watching the war and the greater their worry about the war would be. It was hypothesized that having relatives in Ukraine can lead to heightened anxiety and depression in individuals outside Ukraine due to concerns for their family’s safety and well-being amidst the war. This hypothesis follows the observation that watching identity-relevant news can include secondary traumatization that may cause mental health problems. The concept of secondary trauma is based on two assumptions ([24]). The first assumption is that the individual is part of a structured network that is connected through ties and mechanisms, and the second assumption is that the degree of closeness within the network will contribute to the strength of the secondary trauma. Negative events faced by the group can affect an individual who belongs to the group. Thus, the presence of relatives and friends in the war zone should contribute to greater engagement with the war, which should be associated with heightened distress in those watching the war in another country. It was also hypothesized that denial about the fact that the war is happening would be positively related to anxiety, anger, and worry about the war. This hypothesis follows research findings that denial is often used as a coping mechanism in situations that cannot be altered ([27]) and has been identified as a strong predictor of distress ([1]). Another hypothesis is that women should have greater anxiety and war worry than men. This follows previous research showing that anxiety symptoms are generally more common in women than men ([32]). Lastly, it was hypothesized that self-efficacy would be negatively associated with anxiety and war worry. People with high self-efficacy believe they have the ability to manage prospective situations and exercise influence over them. Research shows that self-efficacy is associated with lower stress ([20]). Therefore, high self-efficacy should be associated with lower anxiety and worry about the war.

## 2. Materials and Methods

### 2.1. Participants and Procedure

Data were collected from April to October 2022, fairly close in time to when the war broke out. Ethics approval was received for each of the national samples, and the participants were university students from Europe and North America. The European sample comprised participants from three countries: Germany, Finland, and the Czech Republic. The North American participants were from Canada and the U.S. The same questionnaire was administered online to all participants in their own language. Table 1 presents the sociodemographic statistics for the five national samples. The average age ranged from 19 to 29 years old. Two-thirds to three-quarters of the samples were female. More participants in North America were in their first year of university studies compared to those in the European samples. All of the data have been uploaded to OSF (https://osf.io/whk48/).

Table 2 presents sociodemographic statistics for participants grouped into two samples: European (N = 769) and North American (N = 467), where the average age ranged from 19 to 29 years old. Almost three-quarters of participants were female in the European sample compared to two-thirds of the sample in North America, and participants were older in the European sample. While most North American participants were in their first year of university, only 27% were in their first year in the European sample.

### 2.2. Variables and Measures

Table 3 presents the study variables and measures of this study. Denial was assessed with a two-item measure from the Brief Cope ([8]). A sample item is “I’ve been refusing to believe that it the war has happened”. Responses ranged from 1, I haven’t been doing this at all, to 4, I’ve been doing this a lot.

Anger when watching the war was measured with 7 items from the Profile of Mood Scale (POMS; [36]). A sample item is “Furious”, and responses ranged from 1, Not at all, to 5, Extremely. Anxiety when watching the war was measured with 6 items from the POMS ([36]). A sample item is “Anxiety”, and responses ranged from 1, Not at all, to 5, Extremely. Self-efficacy was assessed with 4 items that were adapted from the General Self-Efficacy Scale ([20]). Responses ranged from 1, Not at all true, to 4, Exactly true. Participants were asked how much they worry about the war in a single item: “In general, how much do you worry about the war?” Responses ranged from 1, Not at all, to 5, Extremely.

The amount of time spent on social media was assessed with a single item, “In general, how many hours/per week do you spend watching or reading about the war on TV and/or the Internet, your phone, etc. (1 item)”? (Section 2.2, Table 3). Participants’ responses were recorded as the number of hours per week, which was treated as a continuous variable in statistical analyses.

The presence of relatives was measured with a single item: “Do you have relatives, friends, or colleagues in Ukraine or Russia at the present time?” Responses were assessed by asking participants to respond using either 1. Yes or 2. No (1 item).

## 3. Results

### 3.1. Statistical Analysis

Analysis of variance was conducted on several key measures in the theoretical model, i.e., situational factors, emotional reactions, and coping. These analyses examined differences in variables between the European and North American samples. Path analysis was conducted to test the fit of the theoretical model to the data separately in the European and North American samples. The minimum required size of each national sample was determined on a priori power calculations ([31]).

The number of hours per week spent watching the war was significantly higher in Europe than in North America (*F*(1, 1251) = 9.14, *p* < 0.01). Anxiety (*F*(1, 1261) = 13.35, *p* < 0.001), anger (*F*(1, 1261) = 25.77, *p* < 0.001), and war worry (*F*(1, 1261) = 79.83, *p* < 0.001) were also significantly higher in the European sample than the North America one, as was denial (*F*(1, 1261) = 4.13, *p* < 0.05). Self-efficacy was significantly lower in Europe than in North America (*F*(1, 1261) = 6.54, *p* < 0.05). The results of a chi-squared analysis showed that Europeans were more likely to report they had relatives, colleagues, or friends in Ukraine or Russia (*χ*^2^(1) = 28.85, *p* < 0.002, φ = 0.15) (see Table 3).

### 3.2. Testing the Theoretical Model

AMOS version 28 was employed for the estimation of relations among variables, errors, and covariance–variance structures ([3]). Specifically, structural equation modeling (SEM) with maximum likelihood estimation (MLE) was used to test the theoretical model. When fitting SEM models with continuous outcomes, maximum likelihood (ML) is the most commonly used estimation method ([21]). Furthermore, missing data were excluded from the SEM analysis. In consequence, SEM was performed on 1228 participants in total. Among these, 761 belonged to the European sample, and 467 belonged to the North American sample. Path analysis was conducted separately on each of the two samples.

Given the substantial sample size, several fit indices were employed to assess the model’s goodness of fit. A non-significant χ^2^ statistic indicated a strong fit to the data. At the same time, a significant χ^2^ statistic could also result from a large sample size. Additionally, the Comparative Fit Index (CFI) and the Incremental Fit Index (IFI) had to exceed 0.95 and 0.90, while the Root Mean Square Error of Approximation (RMSEA) and the Standardized Root Mean Square Residual (SRMR) needed to be less than 0.08, as suggested by [6] ([6]), [19] ([19]), and [23] ([23]).

### 3.3. European Data

When the model was tested with the European sample, the fit indices indicated an acceptable fit (χ^2^(13) = 87.27, *p* < 0.001), even though the chi-square test was significant, which could be due to the large sample size, and almost all paths were significant (see Figure 2). Furthermore, the CFI (0.922) and IFI (0.923) were satisfactory. The RMSEA (0.087) and SRMR (0.069) indicated an acceptable fit.

An examination of the standardized path coefficients showed that having relatives or friends in the Ukraine or Russia was correlated with more hours spent watching the war. In turn, the number of hours spent watching the war was positively associated with emotional reactions, such as war worry, anger, and anxiety, when watching the war. Emotional reactions, such as anxiety, anger, and war worry, were positively interrelated, as was denial, which was used as a coping strategy. However, self-efficacy as a coping strategy was negatively related to anxiety and war worry when watching the war. Also, anxiety and war worry were higher in women than in men.

### 3.4. North American Data

When the model was tested with the North American sample, the fit indices indicated an acceptable fit of the model to the data (χ^2^(13) = 54.35, *p* < 0.001). The significant chi-squared value could be due to the large sample size, and almost all paths were significant (see Figure 3). The CFI (0.916) and IFI (0.919) were satisfactory. Similarly, the RMSEA (0.083) and SRMR (0.061) indicated an acceptable fit of the model to the data.

An examination of the standardized path coefficients showed that having relatives or friends in the Ukraine or Russia was correlated with more hours spent watching the war and war worry. Furthermore, the number of hours spent watching the war was positively associated with war worry, anger, and anxiety when watching the war. Emotional reactions, such as anxiety, anger, and war worry, were positively interrelated, as was denial. Self-efficacy was negatively related to anxiety but not significantly related to war worry. Gender was not significantly related to anxiety or war worry.

To sum up, the European and North American samples demonstrated an acceptable model fit, with satisfactory values for the goodness-of-fit indices. In each region, having relatives or friends in Ukraine or Russia was associated with more hours spent watching the war, which, in turn, was correlated with higher levels of emotional reactions, including war-related worry, anger, and anxiety. Emotional reactions, such as anxiety, anger, and war-related worry, were positively intercorrelated, and denial emerged as a common coping strategy. Additionally, self-efficacy was negatively related to anxiety in both samples.

Differences were observed in the role of gender and the relationship between self-efficacy and war worry. In the European sample, women reported higher levels of anxiety and war-related worry compared to men, whereas gender was not significantly related to anxiety or war-related worry in the North American sample. Moreover, self-efficacy was not significantly associated with war worry in the North American sample, unlike in the European sample, where it was negatively related to war worry.

## 4. Discussion

Watching the war in Ukraine has generated extensive negative emotional reactions in many people in countries worldwide. This research focused on the study of psychological distress related to the war in two samples of university students: those who lived in Europe and those who resided in North America. Psychological distress manifested as anxiety, anger, and worry about the war. We hypothesized that European participants would be more involved in the war, would watch the war more often on social media and TV, and would experience greater distress. This is due to the greater proximity and similarity of Europeans to the victims of the war, as well as the fact that they have more relatives, friends, or colleagues in the war zone. We also hypothesized that greater concern for these individuals should result in greater psychological distress when watching the war. Therefore, it is not surprising that European participants spent more hours watching the war and experienced greater negative emotional reactions, such as anger, anxiety, and worry, while watching the war than their North American counterparts.

In both samples, the findings showed that the more hours participants spent watching the war, the greater their anxiety, anger, and worry about the war. These results parallel previous research showing that watching violence on social media and TV is associated with psychological distress, even though the individuals are not physically present at the site of the war ([10]). The findings further showed that European participants spent significantly more hours watching the war than North Americans, as expected. Europeans also had higher scores for anger, anxiety, and worry while watching the war than their North American counterparts. Since Europeans were physically closer to the site of the war and had greater ties to the victims of the war, it was expected that they would identify with them more than the North Americans. This would lead to greater distress while watching the war on social media

The theoretical model introduced here, integrating hours watching the war, relatives, friends, or colleagues in both countries, emotional factors, gender, and coping, fits the data in two independent samples. The findings showed that denial was positively related to anger, anxiety, and worry about the war in both samples. These findings corroborate previous research that denial is a maladaptive coping strategy ([37]) and is itself often a symptom of distress. Further results in this study showed that denial was significantly higher in the European sample than in the North American one. Given that the European participants were watching the war for more hours and their psychological distress was greater than that of their North American counterparts, they may have been using denial more as a way of coping with their distress.

The present findings show a significant association between watching depictions of the war on social media and psychological distress, parallel research that showed a relationship between psychological well-being and salience of the war on social media. For example, the results of a longitudinal study of a sample from 17 European countries showed that daily well-being was lower on days when the war was more salient on social media, using the daily number of tweets that contained the keyword Ukraine worldwide, which was used as an indicator of the daily salience of the war on social media ([34]). For example, a decline in well-being on the day of the invasion coincided with an increase in Ukraine-related tweets.

There are some limitations of the present research. First, the data were collected from university students, who may not be representative of the general population, given their youth. Future research could replicate this study using samples that are somewhat older and more heterogeneous while assessing further subgroup differences. At the same time, student populations are not homogeneous, and emotional reactions in subgroups should be assessed in further research. Second, three of the measures were single items, which are usually not as stable as multi-item measures and do not allow for further differentiation. Third, since the data were cross-sectional and collected in 2022, causal interpretations of the relationships among variables are not possible, and interpretations may not be generalizable to further stages of the conflict. Fourth, in this study, participants were asked, in general, how many hours/per week they spent watching or reading about the war on TV and/or the Internet, their phone, etc. The number of hours spent consuming news was used as a proxy for involvement in the war. At the same time, in this study, this variable did not differentiate between the consumption of news via traditional television broadcasts and other kinds of social media platforms. Given the distinct characteristics and documented differential impacts of these two modalities in communication research ([33]), it would be beneficial to incorporate this distinction in future research that studies the relationship between the consumption of media messages and psychological reactions to watching the war. Another limitation of this study concerns the temporal scope of data collection, which occurred in 2022. The war began in 2022 and continues in 2025, which represents a substantial time period since the data were collected for this study. Within the context of the war, this time frame has been marked by significant developments, including increasingly severe events and highly impactful news narratives originating from the conflict zones reported in various media forms. Another limitation is that contextual factors, such as different media coverage, political climate, and campus climate influences, were not assessed but could have affected participants’ emotional reactions and should be considered in future research. At the same time, student populations are not homogeneous, and subgroup differences should be assessed in further research.

When testing the theoretical model, some of the values of the standardized regression paths were low. However, this does not invalidate a path if the relationship is theoretically supported, and it contributes to understanding the structural relations among variables ([26]). Moreover, a low-to-moderate path coefficient does not indicate a limitation, especially when it is statistically significant and based on a theory that aligns with the literature. A small but significant path coefficient indicates that a relationship really exists, even if its magnitude is not large. Given the heterogeneity of the samples, it is not surprising that some of the paths are weaker than others. In many psychological studies, small effects are expected because outcomes are multifactorial. In this context, weak but significant effects may still be meaningful ([14]). In the present study, the effects are robust across different countries; testing across heterogeneous samples typically reduces (cor-)relation magnitudes; so, detecting significant effects across contexts indicates reliable findings. At the same time, the theoretical model had an acceptable fit to the data in both the European and North American samples, consisting of a total of five national samples. More importantly, the fact that the theoretical model fit the data in both samples attests to its validity as a model of distress when watching the war in vastly different samples. Despite the findings that Europeans found the war more distressing and had more relatives in the war zone than North Americans, the model was a good fit in both samples. Furthermore, the data suggest that the hypothesized relationships in the model are strong enough to support the model.

Despite these limitations, the theoretical model put forward here was an acceptable fit to the data in two different continental samples, Europeans and North Americans, and most of the hypothesized relationships among the study variables were found in both samples. Furthermore, the findings reported here have implications for reducing distress associated with watching the war. By increasing self-efficacy, it may be possible to reduce both worry and anxiety while watching the war, thus increasing psychological well-being in populations geographically near and far from the conflict.

## 5. Conclusions

The present study specifies the emotions that were associated with psychological distress while watching the war, namely, anger, worry, and anxiety, thus indicating specific areas to be targeted in interventions that could be taken in the future to help alleviate distress related to watching a war. In addition, this study demonstrated that the presence of relatives, friends, or colleagues in both countries was associated with psychological distress through the number of hours watching the war, thus indicating the importance of incorporating individual factors into research that examines the relationship between depictions of armed conflict on social media and psychological distress

As predicted, self-efficacy had a protective effect against experiencing anxiety. Previous research confirms that having higher levels of self-efficacy decreases a person’s chances of experiencing stress ([39]; [22]). The reason is that self-efficacy increases a person’s sense of being in control of a situation and may thus act as a safeguard against stress. Self-efficacy has been identified as a personal resource that is associated with lower distress. In the present study, self-efficacy was related to lower anxiety while watching the war in both samples. That is, to the extent that individuals possessed higher self-efficacy, they were less likely to experience anxiety when watching the war, a finding that was replicated in both samples, thus adding to the validity and robustness of the finding. Hence, by strengthening self-efficacy, anxiety associated with watching the war could be reduced.

To summarize, the present findings on the relationship between watching armed conflict on social media and psychological distress are unique in three main ways. First, they demonstrate the importance of taking into account national differences in this kind of research since those residing in countries closer to the armed conflict not only experienced more distress but also had more relatives in the countries in conflict. Second, the present results highlight the importance of incorporating individual differences, such as self-efficacy, into this research, since those with higher self-efficacy scores reported significantly lower anxiety than their lower-scoring counterparts. Third, by defining psychological distress in terms of anxiety, anger, and worry while watching the war, the present research specifies those emotional reactions that comprise psychological distress, thus facilitating the design of interventions that could be taken to reduce psychological distress in response to watching violence on the media.

## Figures and Tables

**Figure 1 behavsci-15-01319-f001:**
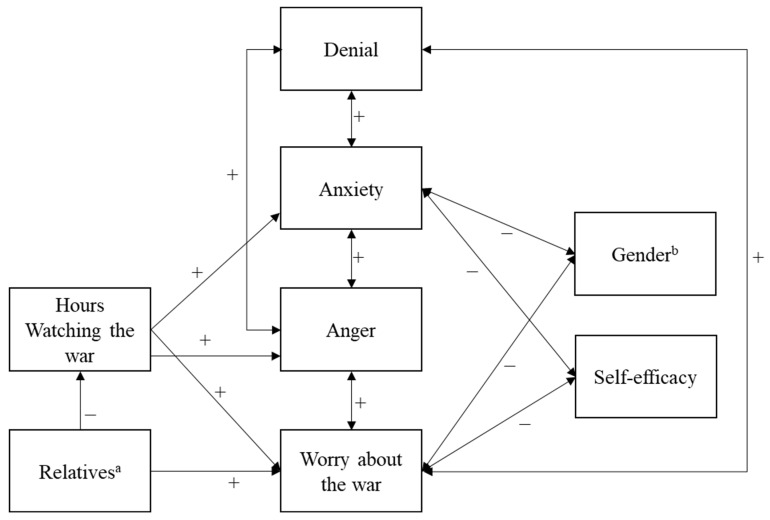
Theoretical model with situational factors, emotional reactions, and coping: Europe and North America. Note: ^a^ Low scores indicate more relatives/friends in Ukraine or Russia. ^b^ 1 = female, 2 = male.

**Figure 2 behavsci-15-01319-f002:**
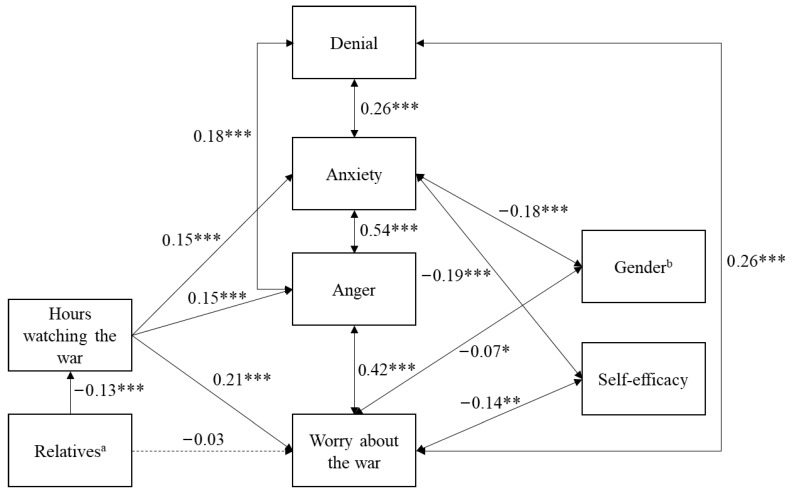
Empirical model with situational factors, emotional reactions, and coping for the European sample. Coefficients in this path analysis are standardized linear regression coefficients. * *p* < 0.05. ** *p* < 0.01. *** *p* < 0.001. Solid lines indicate significant paths, whereas the dashed line indicates a non-significant path. ^a^ Low scores indicate more relatives/friends in Ukraine or Russia. ^b^ 1 = female, 2 = male.

**Figure 3 behavsci-15-01319-f003:**
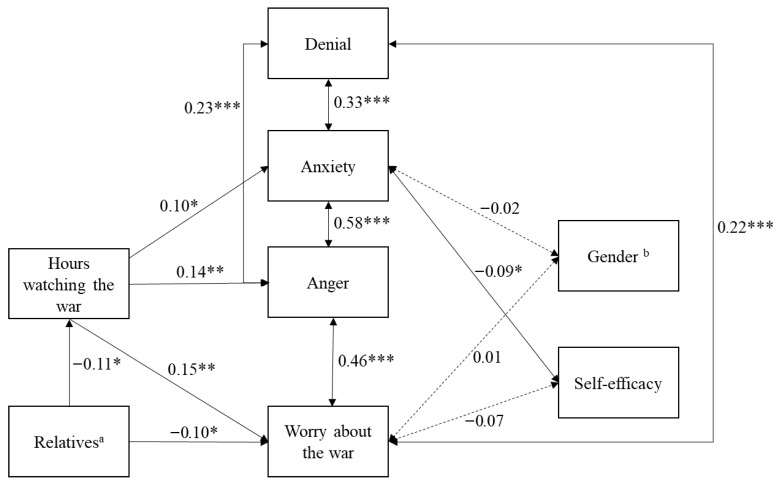
Empirical model with situational factors, emotional reactions, and coping for the North American sample. Coefficients in this path analysis are standardized linear regression coefficients. * *p* < 0.05. ** *p* < 0.01. *** *p* < 0.001. Solid lines indicate significant paths, whereas dashed lines indicate non-significant paths. ^a^ Low scores indicate more relatives/friends in Ukraine or Russia. ^b^ 1 = female, 2 = male.

**Table 1 behavsci-15-01319-t001:** Demographics for the five national samples.

Variable	Germany	Finland	Czech Republic	Canada	U.S.
N	342	213	214	310	157
Age mean (*SD*)	25.46 (8.32)	29.26 (8.64)	23.10 (6.21)	19.85 (3.56)	19.16 (1.69)
% Female	76	65	73	62	73
Student status in %					
First-year student	25	21	37	68	84
Second year or higher	75	79	63	32	16

**Table 2 behavsci-15-01319-t002:** Demographics for the two continental samples.

Variable	Europe	North America
Total N (% Female)	769 (72)	467 (66)
Age mean (*SD*)	25.94 (7.72)	19.50 (2.62)
Student status % First year % Second year or higher	27 73	76 24

**Table 3 behavsci-15-01319-t003:** Descriptive statistics and reliabilities for study variables in the two continental samples.

Measure	Authors	Number of Items	Sample Item	Europe			North America		
				*M*	*SD*	α	*M*	*SD*	α
Denial	[8] ([8])	2	I’ve been refusing to believe that it {the war} has happened	1.89	0.74	0.66	1.80	0.82	0.81
Anger when watching the war	[36] ([36])	7	Using the scale below, indicate your feelings when you watch the war or read about it. “Furious”	2.85	0.97	0.87	2.54	1.08	0.92
Anxiety when watching the war	[36] ([36])	6	Using the scale below, indicate your feelings when you watch the war or read about it. “Anxious”	2.99	0.98	0.87	2.77	0.99	0.90
Self-efficacy	Adapted from [20] ([20])	4	I am confident that I can deal efficiently with unexpected events	2.70	0.66	0.82	2.90	0.54	0.80
War worry	The researchers	1	In general, how much do you worry about the war?	3.44	0.96	-	2.90	1.05	-
# Hours watching the war ^1^	The researchers	1	In general, how many hours/per week do you spend watching or reading about the war on TV and/or the Internet, your phone, etc?	4.62	5.69	-	3.57	6.32	-
Relatives ^2^	The researchers	1	Do you have relatives, friends, or colleagues in Ukraine or Russia at the present time?	1.81	0.39	-	1.92	0.27	-

^1^ A continuous variable. ^2^ The lower the numerical value, the more relatives, friends, or colleagues in Ukraine or Russia.

## Data Availability

Data for this study can be found here: OSF (https://osf.io/whk48/).

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
