# Peer review of "Watching the Russian–Ukrainian War: Comparison Between Europe and North America"

_behavsci, 2025, doi:10.3390/bs15101319_

Round 1

Reviewer 1 Report

Comments and Suggestions for Authors

The manuscript effectively investigates the widespread phenomenon of online news consumption pertaining to the Russian-Ukrainian War. The research endeavors to ascertain the psychological influences and effects of news viewership, particularly among individuals with direct or indirect connections to the conflict, and differentiates findings based on nationality. The paper is generally well-structured and presents a compelling case for publication. However, several reservations warrant consideration:

  1. Introduction and Theoretical Framing: While the introduction appropriately acknowledges the concept of stress, its brevity (potentially due to word count constraints) suggests that a more extensive discussion, particularly concerning the theoretical underpinnings of stress, might be more suitably situated within the theoretical framework section.

  1. Differentiation of Media Platforms: The current analysis lacks a clear differentiation between the consumption of news via traditional television broadcasts and through social media platforms. Given the distinct characteristics and documented differential impacts of these two modalities in communication research, it would be beneficial to incorporate this distinction, both in the methodological design and in the interpretation of findings and conclusions.

  1. Hypothesis Formulation and Referencing: The hypothesis section, as currently formulated, lacks supporting academic citations. In scholarly work, hypotheses should ideally be derived from, and explicitly grounded in, existing theoretical frameworks and empirical findings within the relevant literature, rather than solely representing the authors' unreferenced propositions. Consequently, this section requires robust academic referencing to substantiate its claims.

  1. Temporal Scope of Data Collection: The data collection was concluded in 2022. Acknowledging the inherent temporal lag associated with academic research, writing, and publication processes, the intervening period until the present (approaching 2026) represents a substantial duration. Within the context of the Russian-Ukrainian War, this timeframe has been marked by significant developments, including increasingly severe events and highly impactful news narratives originating from the conflict zones. If the inclusion of more recent data is not feasible, it is imperative that this temporal limitation and its potential implications for the generalizability or contemporary relevance of the findings be explicitly acknowledged within the manuscript.

Author Response

Thank you very much to the Reviewers for taking the time to review our manuscript, Watching the Russion-Ukraine War: Comparison of Europe and North America. We appreciate the reviews which have provided valuable additions to our study which we have incorporated in the manuscript. In this letter we respond to the Reviewers’ comments and explain, point by point, the details of the manuscript revisions. In addition, the revisions are highlighted in the manuscript.

Reviewer 2 Report

Comments and Suggestions for Authors

Title of Manuscript:

Watching the Russian-Ukrainian War: Comparison of Europe 2 and North America

General assessment and comments on this manuscript

The manuscript addresses a highly relevant and timely topic: the psychological consequences of media exposure to the Russian-Ukrainian war but the perception of a conflict is volatile and may a longitudinal approach and follow-up.

This study is valuable in its cross-national scope and relatively large student sample. The findings—showing stronger psychological reactions in European students compared to North American ones, the buffering role of self-efficacy, and the role of personal ties to the conflict—add important empirical evidence to the growing literature on indirect exposure to war and trauma.

At the same time, there are areas where the manuscript could be strengthened, particularly in the theoretical background, the consideration of contextual factors, and the treatment of student populations as heterogeneous groups.

Strengths of this study

Timeliness and importance: the war in Ukraine is an unprecedented global crisis, which hits the world since 2022 and examining how young people experience its psychological impacts through media is crucial.

Cross-Cultural Sample: including data from both Europe and North America allows for meaningful comparisons.

Sample Size: the use of 1260 participants provides statistical robustness.

Methodological Variety: the use of both ANOVA and path analyses shows the authors’ intent to capture both group-level differences and mediating relationships.

Applied Implications: yhe findings may inform interventions to support young people’s mental health during crises and provide an opportunity to shape these interventions.

Areas for improvement

Theoretical Background

The current theoretical framing could be expanded. While the authors predict stronger distress in European students due to geographical and political proximity, additional perspectives could enrich the analysis.

For example:

  • Media exposure theories (e.g., cultivation theory, framing effects, or social amplification of risk) would provide a more solid basis for understanding the role of watching the war.
  • Psychological theories of stress and coping (instead of the old model of Lazarus and Folkman (1984), could help explain why self-efficacy acts as a protective factor. Related concepts such as ‘salutogenesis’, ‘hardiness’, ‘self-determination’, may offer additional insights in order to further improve the theoretical analysis.

This expansion would strengthen the link between hypotheses and findings.

Contextual Influences Beyond Geography

The study treats country of residence as the main contextual variable, but other factors are likely to influence appraisal of the conflict and emotional reactions:

  • Media coverage: differences in media quantity, tone, and framing may strongly influence emotional outcomes.
  • Political climate: the rhetoric of political leaders and the stance of governments toward the war could shape public concern.
  • Campus climate: student activism, demonstrations, and institutional reactions on university campuses could also contribute to how students perceive and emotionally engage with the war.

Acknowledging these factors, even if they cannot be directly measured, would improve the discussion of findings.

Heterogeneity Within Student Populations

The study groups participants by country, but student populations are not homogeneous. For example:

  • Country of origin: students with personal ties to Eastern Europe may react differently than local-born peers, regardless of whether they are in Europe or North America.
  • Cultural subgroups: ethnic, linguistic, or international student status may also moderate the observed effects.

Considering these subgroup differences would provide a more nuanced understanding of the results.

Writing and Precision

  • Some sentences could be shortened or clarified for readability (e.g., "Since European countries have greater ties…" could be rewritten as a hypothesis).
  • Care should be taken with causal language, given the cross-sectional design (e.g., “was indirectly related to” could be rephrased as “was associated with”).

Recommendations

  • Strengthen the theoretical background, drawing from media psychology and coping literature.
  • Acknowledge the role of media coverage, political context, and campus-level factors in shaping reactions.
  • Discuss heterogeneity within student samples, especially regarding international and minority students.
  • Revise the writing for conciseness and precision, and avoid implying causality where not supported.

Overall Recommendation

Major revisions.

This is a valuable study with strong potential for contribution. With a richer theoretical foundation, broader contextual consideration, and attention to within-country diversity, the manuscript will become a much stronger piece of work.

Detailed comments:

Line 74 – check spelling: ‘effects’ instead of ‘effecst’

Variable and Measures:

  • Please provide psychometric variables of the scales used
  • Avoid using abbreviations of scales before having used the full name and origin

(ex. POMS scale)

  • I would also expect some more explanations on how variables such as the amount of time watching social media are computed as variables in the model

Comments on the theoretical model which is offered as a concept under study:

Path analysis is essentially a system of multiple regression equations, where variables are modeled as predictors and outcomes in a theoretically specified structure.

For the model to make sense: 1) the relationships among the variables should be strong enough (statistically and substantively) to justify linking them, and, 2) path coefficients which are derived from the correlations among the observed variables should not bet weak. I am surprised that in this model the bivariate correlations are very low (e.g., < .20), which could mean that the path coefficients also tend to be weak.

In this case almost all correlations among the main constructs are below .20, which may suggest:

- Weak substantive relationships: the variables of this model might not be theoretically or empirically connected in the way the model assumes.

- Model misspecification: the chosen predictors may not be the most relevant ones for explaining the outcomes.

- Measurement issues: the constructs may have been measured poorly (e.g., low reliability, vague items, or constructs too broad).

- Contextual mismatch: the theoretical model may not fit the specific population/sample being studied are not supported by a strong empiral basis.

Two implications for theoretical soundness of this study

- If theory or researchers predict strong associations, but empirically the correlations are weak, this challenges the theoretical validity of the model. It may indicate that the conceptual framework does not align well with the observed data.

- If the theory allows for only weak relationships (because many factors contribute), then low correlations might still be reasonable—but they limit the explanatory power of the model. If this is the case, I would expect this issue to be treated in the limitations section.

In short: consistently low correlations undermine confidence that the theoretical model captures the true underlying processes. It suggests either the theory needs refinement, or alternative constructs need to be considered.

Recommendations to the research team:

- Re-examine the constructs: are the variables chosen theoretically the right ones?

- Check measurement quality: were the scales reliable and valid?

- Test alternative models: could omitted variables explain the outcome better?

- Theoretical reflection: Perhaps the assumed linear causal relationships are oversimplified, and non-linear, moderating, or contextual effects are more appropriate.

Because, if all correlations in a path model are very low (below .20), the theoretical problem is that the assumed relationships may not exist strongly enough to support the model. This casts doubt on the theoretical soundness and suggests the conceptual framework might be weak, misapplied, or require revision.

Finally, a detailed comment on Figure 2.

‘mpirical Model’ -> add ‘E’

Author Response

(The authors gave the same response as above.)

Round 2

Reviewer 2 Report

Comments and Suggestions for Authors

Final Review Report

Manuscript Title: Watching the Russian-Ukrainian War: Comparison of Europe 2 and North America

The revised version of the manuscript demonstrates significant improvement and is now ready for publication.

The authors have clearly addressed most of the comments and recommendations provided on the previous version. Notably, the inclusion of the Lazarus & Folkman stress model and Antonovsky’s theory of salutogenesis has added substantial theoretical depth to the paper. These frameworks not only strengthen the conceptual underpinnings of the study but also clarify the research hypotheses, which are now much more clearly articulated and grounded in solid theoretical reasoning.

Further, the authors have responded well to earlier concerns regarding the use of abbreviations, the rather low correlation relationships and the operationalization of variables. These adjustments contribute to the overall clarity and methodological rigor of the text.

While a number of minor typos, spelling mistakes, and formatting issues remain, they do not detract significantly from the quality of the manuscript and can easily be corrected during the final editorial process.

Overall, the authors are to be congratulated for their thorough revisions and for producing a final version that meets the standards for publication. The manuscript provides valuable insight and is a meaningful contribution to the field.

Recommendation: Accept for publication (minor typographical edits remaining)